# Nusinersen Induces Disease-Severity-Specific Neurometabolic Effects in Spinal Muscular Atrophy

**DOI:** 10.3390/biom12101431

**Published:** 2022-10-06

**Authors:** Francesco Errico, Carmen Marino, Manuela Grimaldi, Tommaso Nuzzo, Valentina Bassareo, Valeria Valsecchi, Chiara Panicucci, Elia Di Schiavi, Tommaso Mazza, Claudio Bruno, Adele D’Amico, Manolo Carta, Anna Maria D’Ursi, Enrico Bertini, Livio Pellizzoni, Alessandro Usiello

**Affiliations:** 1Laboratory of Translational Neuroscience, Ceinge Biotecnologie Avanzate, 80145 Naples, Italy; 2Department of Agricultural Sciences, University of Naples “Federico II”, 80055 Portici, Italy; 3Department of Pharmacy, University of Salerno, 84084 Fisciano, Italy; 4Department of Environmental, Biological and Pharmaceutical Science and Technologies, Università degli Studi della Campania “Luigi Vanvitelli”, 81100 Caserta, Italy; 5Department of Biomedical Sciences, University of Cagliari, 09042 Monserrato, Italy; 6Division of Pharmacology, Department of Neuroscience, Reproductive and Dentistry Sciences, School of Medicine, University of Naples “Federico II”, 80131 Naples, Italy; 7Center of Translational and Experimental Myology, IRCCS Istituto Giannina Gaslini, 16147 Genoa, Italy; 8Institute of Biosciences and BioResources (IBBR), CNR, 80131 Naples, Italy; 9IRCCS Casa Sollievo della Sofferenza, Bioinformatics Unit, 71013 San Giovanni Rotondo, Italy; 10Department of Neuroscience, Rehabilitation, Ophtalmology, Genetics, Maternal and Child Health-DINOGMI, University of Genova, 16132 Genoa, Italy; 11Unit of Neuromuscular and Neurodegenerative Disorders, Department of Neurosciences, Bambino Gesù Children’s Hospital IRCCS, 00163 Roma, Italy; 12Center for Motor Neuron Biology and Disease, Columbia University, New York, NY 10032, USA; 13Department of Pathology and Cell Biology, Columbia University, New York, NY 10032, USA; 14Department of Neurology, Columbia University, New York, NY 10032, USA

**Keywords:** spinal muscular atrophy (SMA), survival motor neuron (SMN), nusinersen, cerebrospinal fluid (CSF), nuclear magnetic resonance (NMR)

## Abstract

Intrathecal delivery of Nusinersen–an antisense oligonucleotide that promotes survival motor neuron (SMN) protein induction–is an approved therapy for spinal muscular atrophy (SMA). Here, we employed nuclear magnetic resonance (NMR) spectroscopy to longitudinally characterize the unknown metabolic effects of Nusinersen in the cerebrospinal fluid (CSF) of SMA patients across disease severity. Modulation of amino acid metabolism is a common denominator of biochemical changes induced by Nusinersen, with distinct downstream metabolic effects according to disease severity. In severe SMA1 patients, Nusinersen stimulates energy-related glucose metabolism. In intermediate SMA2 patients, Nusinersen effects are also related to energy homeostasis but involve ketone body and fatty acid biosynthesis. In milder SMA3 patients, Nusinersen mainly modulates amino acid metabolism. Moreover, Nusinersen modifies the CSF metabolome of a more severe clinical group towards the profile of untreated SMA patients with milder disease. These findings reveal disease severity-specific neurometabolic signatures of Nusinersen treatment, suggesting a selective modulation of peripheral organ metabolism by this CNS-directed therapy in severe SMA patients.

## 1. Introduction

Spinal muscular atrophy (SMA) is caused by homozygous deletions or mutations in the *survival motor neuron 1* (*SMN1*) gene, resulting in reduced expression of the SMN protein, which leads to the progressive degeneration of motor neurons and atrophy of skeletal muscle [1,2]. SMA patients display a wide range of clinical manifestations and are classified into three main groups (types 1, 2, and 3), according to the age of onset and maximum motor function achieved [1,2,3]. Disease severity inversely correlates with the levels of *SMN* expression and the number of copies of *SMN2* [1,2]—the *SMN1* paralogue gene. Notably, a single base pair difference between *SMN1* and *SMN2* accounts for the decreased efficiency of exon 7 inclusion into mature transcripts from the *SMN2* gene [1,2]. Therefore, *SMN2* produces only a fraction of full-length SMN protein, as compared to *SMN1*, and cannot fully compensate for its loss in SMA. Since the *SMN2* copy number is the primary genetic modifier of disease severity, several therapeutic approaches for SMA have focused on increasing expression of SMN through modulation of *SMN2* exon 7 splicing [4,5,6].

Remarkable advances in the development of SMA therapeutics have led to the recent approval of three distinct SMN-inducing treatments for SMA patients [7,8,9]. These therapies include an antisense oligonucleotide (Nusinersen) and a small molecule (Risdiplam) that directly modulate *SMN2* splicing, to promote the inclusion of exon 7 [10,11,12,13,14], as well as SMN replacement by gene therapy with an AAV9 vector (onasemnogene abeparvovec-xioi) [15,16,17]. These therapies differ in their mechanisms of action, mode of administration, and the specific patient population for which they have been approved [7,8,9]. 

Nusinersen—approved by the FDA in 2016 as the first treatment for infants, children, and adults with SMA across the severity spectrum—is currently the most widely diffused SMA therapy, with more than 11,000 patients treated worldwide [7,8,9]. It is also the only treatment restricted to the central nervous system (CNS) of SMA patients, who receive repeated intrathecal injections of this *SMN2* splicing–modifying antisense nucleotide, consisting of four loading doses in the first two months of treatment and one maintenance dose every four months afterward [11,12,13]. Clinical trials and real-world data have demonstrated remarkable beneficial effects of treatment with Nusinersen on the motor function when administered presymptomatically or at an early stage of the disease in severe SMA infants [11,12,18,19], while the treatment was much less efficacious when delivered at later stages of the disease course [20,21]. Nusinersen also produces significant but relatively modest clinical benefits in patients with milder forms of the disease [13,22]. 

It is now widely accepted that neither Nusinersen nor other SMA therapies represent a cure for the disease [7,8,9]. Incomplete correction of disease symptoms combined with variability in the clinical response to treatment represent unmet needs that remain to be addressed. Time of intervention has emerged as a key factor that influences the efficacy of current therapies, with earlier treatment being most effective and delayed treatment often encountering a more muted clinical response [7,8,9]—a scenario that is also consistent with a recent report showing that motor axon development deficits occur prenatally in severe SMA patients [23]. Another limitation is the lack of therapeutics that could enhance the clinical benefit of SMN-inducing drugs through combinatorial treatment [24]. In this context, the identification of neurometabolic signatures that correlate with disease severity and accurately reflect clinical improvement or the lack thereof by current therapies is critical, not only to explain differences in clinical response but also to guide the development of new therapeutics. However, we have a limited understanding of the biochemical abnormalities associated with SMA pathology and the metabolic effects of SMN-inducing therapies in the CNS of patients. Moreover, despite increasing efforts to identify the biomarkers of disease progression and therapeutic efficacy [25,26,27,28,29,30], there has been a paucity of metabolomic studies focused on the cerebrospinal fluid (CSF) of SMA patients [31], which would have the unique potential to provide direct insights into the disease-associated neurochemical disturbances as well as the therapy-induced changes of CNS-related metabolism, both of which are unknown.

Here, we sought to address these issues by determining the metabolic effects of Nusinersen treatment in the CSF, the biofluid to which Nusinersen is delivered in order to drive its therapeutic effects through SMN induction. To do so, we used nuclear magnetic resonance (NMR) spectroscopy to longitudinally profile the CSF from SMA patients across the spectrum of disease severity before and after Nusinersen treatment. Our findings demonstrate that Nusinersen induces profound, yet distinct, neurometabolic changes in SMA patients of varying disease severity.

## 2. Materials and Methods

### 2.1. Patients and Samples Characteristics

This two-center study was conducted on 27 patients affected by SMA1 (*n* = 12), SMA2 (*n* = 7), and SMA3 (*n* = 8), who received intrathecal treatment with Nusinersen at the Bambino Gesù Hospital (Rome, Italy) and at the Giannina Gaslini Institute (Genoa, Italy). The study was approved by the local ethics committees of Bambino Gesù Hospital (2395_OPBG_2021) and Giannina Gaslini Institute (2395_OPBG_2021). All participants and/or their legal guardians signed a written informed consent. All patients were clinically diagnosed and genetically confirmed, and their SMN2 copy numbers were also determined. All SMA1 patients, irrespective of their age and disease severity, were part of the Expanded Access Programme (EAP) for compassionate use for patients with the infantile form only, which occurred in Italy between November 2016 and November 2017. The overall clinical response of these patients to Nusinersen treatment has previously been reported as part of the full Italian cohort and showed that therapeutic efficacy is related to age and clinical severity at baseline [21,32]. SMA2 and SMA3 patients have also been reported previously [22]. Demographic characteristics and clinical data of patients are reported in Table 1. All 27 patients received injections of Nusinersen, per standard protocol. For the purpose of this study, only CSF samples collected at day 0 (T0; baseline), day 64 (T1; after 3 Nusinersen injections), and day 302 (T2; after 5 Nusinersen injections) were evaluated by NMR analysis. Of the 81 lumbar punctures collected, CSF analysis of one lumbar puncture at T2 was not available for pH or total protein measurements, due to the small volume. CSF pH and total protein content did not statistically differ among untreated patients with different disease severity (SMA1 vs. SMA2 vs. SMA3, pH *p* = 0.704, total proteins: *p* = 0.845; Kruskal–Wallis test) and at the different time points of Nusinersen administration (SMA1: T0 vs. T1 vs. T2, pH *p* = 0.469, total proteins *p* = 0.761; SMA2: T0 vs. T1 vs. T2, pH *p* = 0.437, total proteins *p* = 0.824; SMA3: T0 vs. T1 vs. T2, pH *p* = 0.221, total proteins *p* = 0.925; Kruskal–Wallis test). Median values with ranges of pH and total protein in CSF are shown in Appendix A. 

### 2.2. Clinical Evaluation

Assessment of patients was performed at baseline before the initiation of treatment (T0), at day 60 (T1), and at day 302 (T2). During each visit, extensive clinical examination was performed by experienced child neurologists or pediatricians with expertise in the SMA field, and anthropometric measurements and vital parameters were collected. Moreover, patients’ feeding status (oral nutrition, naso-gastric tube (NG), or percutaneous gastrostomy), nutritional status (postulated by body mass index), and respiratory function (spontaneous breathing, non-invasive ventilation (NIV), or tracheostomy) were recorded. All 12 SMA1 patients were older than 5 months at the beginning of treatment, with age ranging from 6 months to 7 years and 8 months. Overall, 6 patients had tracheostomy, 4 were under NIV for <16 h/day, and 2 patients showed spontaneous breathing. In addition, 10 patients had gastrostomy, and, in all patients, the BMI fell into the underweight range (<18). The age of the 7 SMA2 patients included in this study ranged from 1.4 years to 12.5 years at baseline. Three of these patients were under NIV, none had gastrostomy, and four patients had a BMI fall below 18. Regarding the 8 SMA3 patients, 3 were ambulant at baseline evaluation, 1 was under NIV for <16 h/day, and none had gastrostomy. The BMI fell below 18 in 3 patients. At T0, T1, and T2, all patients were assessed using standardized motor function tests, chosen according to their age and motor function. Functional assessments were performed by expert physiotherapists trained with standardized procedure manual [33] and reliability sessions. SMA1 patients were assessed with CHOP-INTEND [34,35], a functional scale including 16 items that is aimed to assess motor function in weak infants. Each item is scored from 0 to 4 (with 0 being no response and 4 being complete level of response), with a total score ranging from 0 to 64. SMA2 and SMA3 patients were evaluated with the HFMSE [35,36], a scale of 33 items investigating the child’s ability to perform different activities. The total score can range from 0, if all the activities are failed, to 66, indicating better motor function. All patients were not wearing spinal jackets or orthoses during the evaluations. 

### 2.3. Intrathecal Treatment with Nusinersen

Intrathecal administration of 12 mg of Nusinersen was performed under hospital environment. A fasting less than 4 h was planned in advance to the procedure in SMA1 patients. During this period, SMA1 patients received intravenous hydration with 5% glucose solution. The time elapsed between the last meal and the lumbar puncture was 6–8 h in SMA2 and SMA3 patients. In SMA1 the procedure was carried out without sedation, whereas for SMA2 and SMA3 patients a sedation with midazolam was applied. No severe adverse events were reported. After the infusion, all patients were recommended to lie down for 2 h to avoid any possible post-lumbar-puncture symptoms.

### 2.4. CSF Sample Collection

CSF samples were collected at the intrathecal administration of Nusinersen in polypropylene tubes and stored at −80 °C until further analysis. Determination of pH and total protein content (Appendix A) and ^1^H-NMR analysis were performed on the CSF sample of each patient. Proteins determination was carried out using the Bradford assay (Bio-Rad, München, Germany) [37].

### 2.5. NMR Sample Preparation and Spectra Acquisition

CSF samples were collected according to standard operating procedures for metabolomic NMR analysis [38]. To 80 μL of CSF were added 420 μL of buffer (50 mM di Na_2_HPO_4_, 1 mM of TSP-d4 and 50 μL of D_2_O), which were transferred into 5 mm NMR tubes for ^1^H-NMR detection. TSP-d4 0.1% in D_2_O was used as an internal reference for the alignment and quantification of the NMR signals [39]. NMR experiments were acquired on a Bruker-AV II 600 MHz spectrometer equipped with a 5 mm triple-resonance z-gradient CryoProbe (Bruker Co, Rheinstetten, Germany at 300 K.) [40]. Topspin version 3.0 was used for spectrometer control and data processing (Bruker Biospin). Moreover, 1D NOESY experiments were acquired using a spectral width of 14 ppm, 16 k data points, excitation sculpting for water suppression, 192 transients, 4 s relaxation delay, and 60 ms mixing time [41]. A weighted Fourier transform was applied to the time domain data, with a 0.5 Hz line-broadening followed by manual phase and baseline correction in preparation for targeted profiling analysis. Resonance assignment, performed with CHENOMIX software [42] on 1D ^1^H CPMG NMR spectra obtained from the CSF polar extracts of SMA patients, detected the presence of 35 metabolites (Figure 1A). The quantification of assigned metabolites was carried out using automated Bayesil software [43].

### 2.6. Statistics

Multivariate statistical analysis and partial least-squares discriminant analysis (PLS-DA) were carried out with MetaboAnalyst 5.0 [44]. Multivariate statistical analysis was carried out on NMR data set to identify the CSF metabolomic profile of SMA patients before and after the Nusinersen treatment. After normalization by sum and Pareto scaling, the data matrices were analyzed by multivariate supervised PLS-DA method [45]. The performance of the PLS-DA model was evaluated using the coefficient Q2 (using the 7-fold internal cross-validation method) and the coefficient R2, defining the variance predicted and explained by the model, respectively. In each cross-validation step, the expected data was compared with the original data, and the squared sum of errors was calculated. The prediction error was then summed up on all samples (predicted residual sum of squares (PRESS)). The PRESS was divided by the initial sum of squares for greater accuracy and subtracted from 1 to resemble the R2 scale [46]. The metabolites that discriminate the cluster detected by the PLS-DA score plot are represented in the loading plot. These metabolites were ranked according to their variable influence on projection (VIP) scores, representing the weighted sums of the PLS-DA weights’ squares, indicating the importance of the variable [47]. Metabolites characterized by VIP > 1 are considered good classifiers between two clusters [47]. The analysis of the KEGG pathways was carried out using the Enrichment tool of MetaboAnalyst. Only pathways with FDR-adjusted *p* values lower than 0.05 and hit values, namely the number of metabolites belonging to the pathways, greater than 1 were considered. The commonality among pathways was calculated and represented using the *UpSetR* ver. 1.4 R package [48]. Back-to-back bar plots of shared common pathways and metabolites were drawn using the *ggplot2* ver. 3.3.5 R package. VENN diagrams were drawn using the web tool *jvenn* [49]. The hierarchical clustering analysis was carried out using MetaboAnalyst 5.0 [50]. The data matrix containing the concentrations of 35 metabolites for 81 samples (12 SMA1 at 3 times, 7 SMA2 at 3 times, and 8 SMA3 at 3 times) was normalized by log and Pareto Scaling. Clustering was performed using the Ward’s method and the distance between the clusters was calculated based on the Euclidean distance [51]. Correlation analyses between the concentration of metabolites and CHOP-INTEND, HFMSE, ΔCHOP-INTEND, or ΔHFMSE score were performed by Pearson coefficient through MetaboAnalyst 5.0. A Person coefficient (r) ≥ 0.70 was used as threshold. 

## 3. Results

### 3.1. NMR Analysis Reveals Disease Severity-Specific Metabolomes in the CSF of Untreated SMA Patients

We performed a retrospective NMR-based metabolomic analysis of CSF samples from untreated SMA patients across the disease-severity spectrum. Specifically, 12 SMA1, 7 SMA2, and 8 SMA3 patients were included in this real-world study (Table 1). Within our cohort of patients, differences in age were significantly found only between SMA1 and SMA3 patients (SMA1 vs. SMA2, *p =* 0.272; SMA1 vs. SMA3, *p =* 0.037; SMA2 vs. SMA3, *p =* 0.165; Mann–Whitney test), while gender did not differ among groups (χ^2^ = 1.107, *p* = 0.575). Metabolomic analysis of CSF samples was performed using 600 MHz ^1^H-NMR spectroscopy, which reliably identified 35 distinct metabolites (Figure 1A). 

Importantly, partial least-squares discriminant analysis (PLS-DA) showed a significant segregation among the CSF metabolomes of the SMA1, SMA2, and SMA3 groups (Figure 1B). In particular, we observed a stronger separation of severe SMA1 from milder SMA2 and SMA3 forms that, in turn, appear closer to each other. To identify the specific molecules responsible for such metabolome discrimination, we performed a variable importance in projection (VIP) score analysis (Figure 1C). The discriminant CSF analytes (VIP > 1) include succinate, pyroglutamic acid, pyruvic acid, ketone bodies (acetoacetate and 3-hydroxybutyrate), and diverse amino acids, including phenylalanine, tyrosine, tryptophan, serine, threonine, glutamine, and isoleucine (Figure 1C). Consistent with this, pathway enrichment analysis revealed a disease-severity-dependent modulation of the biochemical pathways associated with energy (ketone body metabolism, Warburg effect, and citric acid cycle) and amino acids metabolism (phenylalanine and tyrosine metabolism, glutamate metabolism, valine, and leucine and isoleucine metabolism) (Figure 1D).

To further dissect and quantify the metabolic differences occurring in the CSF of naive SMA1, SMA2, and SMA3 patients, we generated a hierarchical heat map showing the abundance of the metabolites with VIP > 1 in the three severity groups (Figure 1E). The tree diagram at the top demonstrated the hierarchical separation of the metabolite concentrations of SMA1 from those of SMA2 and SMA3 (Figure 1E), pointing to greater metabolic differences in the CSF from the most severe disease form, compared to the milder ones. Among the metabolites specifically reduced in SMA1 patients, we found glucose, succinate, and citric acid, which overall play a key role in glycolysis and citric acid cycle. Conversely, NMR analysis documented higher levels of creatinine in milder SMA2 and SMA3 patients, compared to SMA1 patients (Figure 1E). This extends to the CSF the identification of creatinine as a candidate biomarker for disease severity and progression, which was previously reported in studies of serum and urine [25,31]. Additionally, increased levels of various amino acids, including lysine, threonine, isoleucine, serine, glutamine, tryptophan, tyrosine, and phenylalanine, as well as the ketone bodies acetoacetate, 3-hydroxybutirate, and acetone, are found in the CSF of SMA1 patients, relative to SMA2 and SMA3 patients (Figure 1E).

### 3.2. NMR Analysis Does Not Identify Nusinersen-Dependent Effects in a Pooled Cohort of SMA Patients with Different Disease Severity

After investigating the basal metabolome profiles of untreated SMA patients, we performed a longitudinal NMR-based metabolomic analysis of CSF samples from SMA patients across the disease-severity spectrum who underwent treatment with Nusinersen. 

CSF samples were collected from all SMA patients at the time of the first Nusinersen injection (T0, corresponding to the time point at which we performed the basal metabolomic analysis described above) as well as 64 days (T1) and 302 days (T2) later, at the time of the fourth and sixth scheduled injections (Figure 2A), which correspond to the loading and maintenance phases of the drug administration protocol, respectively.

The clinical response to Nusinersen was evaluated at 302 days of treatment using the Children’s Hospital of Philadelphia Infant Test of Neuromuscular Disorders (CHOP-INTEND) for SMA1 patients and the Hammersmith Functional Motor Scale Expanded (HFMSE) for SMA2 and SMA3 patients (Table 1 and Appendix A). We first sought to analyze the NMR results obtained from the CSF of all SMA patients as a whole, irrespective of their disease severity. PLS-DA showed that the CSF metabolomic profile of the whole cohort of SMA1-3 patients, either at T1 or at T2, was not significantly different from that of the same patients before Nusinersen therapy (Figure 2B,C and Appendix A). Consistent with previous work [31], our NMR analysis did not identify significant metabolic changes induced by Nusinersen treatment in the CSF of SMA patients when pooled into a single cohort, irrespective of their clinical phenotypes.

### 3.3. Nusinersen Modulates Amino Acids and Glucose Metabolism in the CSF of SMA1 Patients

The lack of identifiable biochemical changes induced by Nusinersen in the whole cohort of SMA patients implies either that this drug exerts no major metabolic effects in the CSF or that its effects are disease-severity-specific, which could be masked in a pooled analysis, when also considering the metabolomic differences at the baseline (Figure 1). To address this, we analyzed the CSF of SMA patients as separate cohorts, according to their clinical severity. First, we performed longitudinal analysis of the CSF of SMA1 patients following short-term (T1) and long-term (T2) administration of Nusinersen, relative to the baseline (T0). Remarkably, PLS-DA analysis revealed a significant separation in the CSF metabolomes of SMA1 patients at T0 and T1 (Figure 3A and Appendix A). 

We then performed a VIP score analysis that revealed various discriminating molecules ranging from glucose and pyruvic acid to amino acids (phenylalanine, tryptophan, isoleucine, valine, tyrosine, glutamine, and leucine) as well as dimethylamine, 2-hydroxyisovaleric acid, formate, acetic acid, and creatine (Figure 3B). Through pathway enrichment analysis, we found that Nusinersen significantly modulates bioenergetic pathways involving glucose metabolism (Figure 3C and Appendix A). We also found the metabolism of amino acids (phenylalanine and tyrosine metabolism and valine, leucine, and isoleucine degradation) and sphingolipids as prominent pathways modulated by Nusinersen at T1.

PLS-DA analysis also showed significant separation between the metabolomic profiles of CSF from SMA1 patients before treatment (T0) and at the maintenance phase (T2) of Nusinersen therapy (Figure 3D and Appendix A). VIP score analysis identified discriminating molecules involved in energy-related processes at T2 that were either the same as (glucose and pyruvic acid) or distinct from (fructose and succinate) those identified at T1 (Figure 3E). Amino acids (histidine, threonine, valine, and glutamine) and the amino acid derivative pyroglutamic acid, together with ketone bodies (acetone, acetoacetate, and 3-hydroxybutyrate), were also found as the most discriminant metabolites between T0 and T2 (Figure 3E). Similar to the loading phase (T1), pathway enrichment analysis highlighted the influence of Nusinersen on biochemical processes directly associated with energy homeostasis (citric acid cycle, carnitine synthesis, and ketone body metabolism) and amino acid metabolism (arginine and proline metabolism, and glutamate metabolism) at the maintenance phase relative to the baseline (Figure 3F and Appendix A). Lastly, PLS-DA analysis showed that the CSF metabolomes of SMA1 patients at T1 were not significantly different from those at T2 (Appendix A), pointing to longitudinal conservation of the main neurometabolic effects of Nusinersen in the CSF of SMA1 patients. 

Finally, we generated a hierarchical heat map showing the abundance of the metabolites with VIP > 1 at T1 and T2 relative to the baseline (Figure 3G). The tree diagram at the top demonstrated hierarchical separation of the metabolite concentrations at T1 and T2 from those at T0 (Figure 3G), confirming a coherent, longitudinal modulation of the CSF metabolome of SMA1 patients by Nusinersen. Of the 23 discriminant metabolites identified, we found that the CSF concentration of 16 metabolites was reduced, while that of 7 metabolites was increased by the Nusinersen treatment (Figure 3G). Notably, many of the discriminating metabolites increased by Nusinersen (fructose, lactic acid, glucose, and pyruvic acid) are directly related to glucose metabolism (Figure 3G). Conversely, ketone bodies and all the amino acids identified by the hierarchical clustering except histidine were decreased (Figure 3G).

Altogether, these results identify a significant impact of Nusinersen on biochemical pathways related to energy homeostasis and amino acid metabolism in the CSF of SMA1 patients. 

### 3.4. Nusinersen Modulates Amino Acid and Ketone Body Metabolism in the CSF of SMA2 Patients

We next performed NMR-based metabolomic analysis of CSF from Nusinersen-treated SMA2 patients. PLS-DA score analysis showed a significant separation between the metabolomic profiles of SMA2 patients between T0 and T1 (Figure 4A and Appendix A).

Interestingly, VIP score analysis identified various amino acids (isoleucine, lysine, leucine, alanine, glutamine, valine, methionine, and threonine) and ketone bodies (acetone, 3-hydroxybutyrate, and acetoacetate) as well as fructose, glycerol, and 2-hydroxybutyric acid among the discriminating metabolites (Figure 4B). Accordingly, pathway enrichment analysis identified ketone body and branched-chain amino acids metabolism (valine, leucine, and isoleucine degradation) as well as fatty acid biosynthesis as the main pathways modulated by Nusinersen at loading phase relative to the baseline (Figure 4C and Appendix A). PLS-DA analysis also showed significant separation of the CSF metabolomes from SMA2 patients at T0 and T2 (Figure 4D and Appendix A), and VIP score analysis identified ketone bodies together with amino acids (tyrosine, histidine, and phenylalanine), xanthine, fructose, 2-hydroxyisovaleric acid, and choline as discriminating molecules (Figure 4E). Consistent with the results at the loading phase, pathway enrichment analysis identified significant Nusinersen-dependent modulation of ketone body and amino acid metabolism (phenylalanine and tyrosine metabolism and tyrosine metabolism) as well as fatty acid biosynthesis at the maintenance phase relative to the baseline (Figure 4F and Appendix A). As for SMA1 patients, we did not find a significant separation between the metabolic effects of Nusinersen in the CSF of SMA2 patients between T1 and T2 (Appendix A). Interestingly, fewer pathways were significantly influenced by Nusinersen in SMA2 patients as compared to SMA1 patients, consistent with more profound metabolic effects in the most severe form of the disease. 

We then performed a hierarchical clustering analysis of SMA2 patients at T1 and T2 relative to the baseline (Figure 4G), which evidenced 20 discriminating molecules with VIP > 1. The resulting heat map revealed hierarchical clustering of both Nusinersen-treated groups away from T0, consistent with significant therapy-induced metabolic changes at both the loading and maintenance phases. The concentration of seven discriminating metabolites was decreased, while that of 13 metabolites, including the ketone bodies, acetone, 3-hydroxybutyrate, and acetoacetate, was increased following treatment with Nusinersen (Figure 4G). 

These results reveal that Nusinersen affects energy homeostasis in the CSF of SMA2 patients through different biochemical pathways compared to SMA1 patients. Specifically, while glucose metabolism is the key pathway modulated in SMA1 patients, the bioenergetic boost is mainly driven by the enhanced ketone body and fatty acid metabolism in Nusinersen-treated SMA2 patients. 

### 3.5. Nusinersen Induces Time-Dependent Changes of Amino Acid Metabolism in the CSF of SMA3 Patients

To investigate the longitudinal metabolic effects of Nusinersen in the milder form of the disease, we performed PLS-DA analysis of NMR data from the CSF of SMA3 patients. This revealed a significant separation between the metabolomic profiles of SMA3 patients before (T0) and after Nusinersen administration at T1 (Figure 5A and Appendix A).

VIP score analysis identified several amino acids (glutamine, tyrosine, methionine, threonine, serine, and valine) among the most discriminant metabolites (Figure 5B). Accordingly, pathway enrichment analysis showed that Nusinersen treatment mainly affected the pathways involved in amino acid metabolism (tyrosine metabolism, methionine metabolism, aspartate metabolism, and glutamine metabolism) and ammonia recycling at the loading phase relative to the baseline (Figure 5C and Appendix A). CSF metabolomic profiles of untreated and Nusinersen-treated SMA3 patients at T2 were also significantly different according to PLS-DA analysis (Figure 5D and Appendix A), and amino acids (histidine, serine, lysine, glutamine, threonine, phenylalanine, tyrosine, and leucine) and ketone bodies (3-hydroxybutirate and acetoacetate) were identified as the most discriminating metabolites (Figure 5E and Appendix A). Similar to the analysis at the loading phase, pathway enrichment confirmed Nusinersen-dependent modulation of amino acid metabolism (glycine and serine metabolism and phenylalanine and tyrosine metabolism) and ammonia recycling at the maintenance phase relative to the baseline (Figure 5F and Appendix A). The metabolomic profiles of CSF were also significantly different between T1 and T2 in SMA3 patients (Appendix A), and VIP score analysis identified various amino acids (tyrosine, glutamine, threonine, and lysine) as well as myoinositol, pyroglutamic acid, and 3-hydroxybutyrate as the metabolites responsible for their discrimination (Appendix A). Moreover, unlike the analysis in SMA1 and SMA2 patients (Figure 3G and Figure 4G), hierarchical clustering of the 17 discriminating metabolites (VIP > 1) identified in SMA3 patients does not separate the two Nusinersen-treated groups away from the untreated group (Figure 5G), underscoring the metabolic differences between the CSF of SMA3 patients at T1 and T2. 

Together, these results indicate that the neurometabolic effects of Nusinersen impact fewer biochemical pathways in milder SMA3 patients, compared to more severe SMA1 and SMA2 patients (Appendix A). The effects in SMA3 patients are predominantly centered around amino acid metabolism, which, thus, emerges as a common feature of Nusinersen treatment in SMA patients across the spectrum of disease severity. However, enhancement of the energy-related pathways found in the CSF of SMA1 and SMA2 patients is not among the effects of Nusinersen in SMA3 patients.

### 3.6. Comparative Analysis of Nusinersen-Dependent Neurometabolic Changes and Disease Severity

We investigated whether Nusinersen would modify the CSF metabolome of a more severe clinical group towards the profile characteristic of untreated SMA patients with a milder clinical condition. To do so, we performed an unbiased hierarchical clustering of the CSF metabolomes from SMA1 patients at T0 and T2 as well as SMA2 patients at T0. Note that neither age nor gender is significantly different among the comparison groups (Table 1). Interestingly, the metabolomic profile of treated SMA1 patients at T2 clustered with that of untreated SMA2 patients, rather than that of SMA1 patients at T0 (Figure 6A). 

Of note, there was a broad similarity in the CSF levels of metabolites related to energy metabolism, such as glucose, succinate, and pyruvic acid, between SMA1 patients at T2 and SMA2 patients at T0, which were higher than their baseline in untreated SMA1 patients. These variations were paralleled by reduced ketone bodies levels in Nusinersen-treated SMA1 patients at T2 and in untreated SMA2 patients at T0 relative to SMA1 patients at T0. Similar hierarchical clustering was performed with the CSF metabolomes from SMA2 patients at T0 and T2 as well as SMA3 patients at T0 (Figure 6B), which did not significantly differ for age or gender (Table 1). Importantly, we found that the CSF profile of Nusinersen-treated SMA2 patients at T2 clustered with that of untreated SMA3 patients, rather than with that of SMA2 patients at the baseline. Thus, Nusinersen changes the CSF metabolomes of SMA1 and SMA2 patients towards the profile of untreated SMA2 and SMA3 patients with respectively milder forms of the disease, consistent with a beneficial metabolic effect of drug treatment. 

Finally, we analyzed whether the metabolites identified by VIP > 1 within each disease severity cohort of Nusinersen-treated patients (see Section 3.3, Section 3.4, Section 3.5) correlated with the CHOP-INTEND or HFMSE scores before (T0) and after (T2) therapy with Nusinersen as well as with the ΔCHOP-INTEND or ΔHFMSE scores (i.e., score post-treatment – score pre-treatment) (Appendix A). Interestingly, we found a negative correlation between the levels of 3-hydroxybutyrate and the CHOP-INTEND scores at T2 in SMA1 patients (Appendix A). Conversely, we found a positive correlation of 3-hydroxybutyrate and alanine with HFMSE at T0 in SMA2 patients (Appendix A). Lastly, the CSF levels of valine positively correlated with HFMSE scores at T2 in SMA3 patients (Appendix A). However, we did not identify any metabolites specifically associated with Nusinersen-dependent motor improvement in SMA patients, as no significant correlations were found with the ΔCHOP-INTEND or ΔHFMSE scores.

## 4. Discussion

The clinical benefits and limitations of disease-modifying SMA therapies such as Nusinersen have emerged from both clinical trials and real-world data [7,8,9,24], highlighting their varying efficacy in ameliorating disease manifestations and the greatest benefit with early intervention. Identifying biochemical changes induced by these treatments is critical to explain the observed differences in clinical response and guide the development of combinatorial therapies that could address the unmet needs of SMA patients. In this context, however, there has been a scarcity of studies investigating the biochemical CSF abnormalities associated with SMA pathology and the neurometabolic effects of SMN-inducing drugs in SMA patients. Here, we used NMR-based approaches to compare the basal CSF metabolome of SMA patients across the spectrum of clinical severity and to determine the neurometabolic impact of Nusinersen at different stages of therapy. 

We show that the basal CSF metabolomes differ according to disease severity in SMA patients, possibly reflecting their distinct clinical manifestations. The observed differences are unlikely due to age variations, as these are statistically significant only between our cohorts of SMA1 and SMA3 patients. Importantly, the discrimination of metabolomes among the three types of SMA is based on differences in pathways associated with energy homeostasis (ketone body metabolism, Warburg effect, and citric acid cycle) and amino acid metabolism (phenylalanine and tyrosine metabolism, glutamate metabolism, valine, leucine and isoleucine metabolism). Moreover, our analysis indicates that creatinine levels in the CSF inversely correlate with SMA severity, further supporting this analyte as a candidate biomarker [25,31]. Altogether, these findings underscore the power of NMR-based metabolomic approaches to capture disease-relevant biochemical changes in SMA patients. However, additional studies in larger cohorts of SMA patients and healthy pediatric controls are necessary to further characterize the link between neurometabolic changes and disease severity emerging from our analysis of naive SMA patients.

Our findings reveal that Nusinersen induces profound changes in the CSF metabolome of SMA patients and identify a set of biochemical signatures that are disease-severity-specific. Thus, this study provides new insights into the metabolic effects of SMN induction in the CNS of SMA patients brought about by Nusinersen treatment, which advance our knowledge of the mechanisms of its therapeutic action. We show that this drug induces a broad range of CSF metabolic changes in all SMA types and that the effects are linked to the severity of the disease. Stratification of patients based on their clinical type is indeed key for the identification of neurometabolic changes. Accordingly, biochemical differences induced by Nusinersen do not emerge if NMR-based analysis of CSF metabolomes is performed using pooled data from SMA patients of different severity (Figure 2), which was the approach used in a previous study [31]. On the contrary, we successfully identified both shared and unique biochemical signatures of Nusinersen treatment by NMR analysis of stratified SMA patients. 

We found that Nusinersen triggers a prominent boost of glucose-related metabolism in SMA1 patients (Figure 3). Accordingly, heatmap analysis shows that the levels of glucose, lactic acid, and pyruvic acid increase in the CSF of SMA1 patients following treatment, consistent with the induction of intermediary metabolism [52]. In agreement with direct involvement in energy homeostasis, Nusinersen also stimulates degradation of branched-chain amino acids, such as valine, leucine, and isoleucine, which can produce acetyl-CoA, and their breakdown likely contributes to the observed increase in the transfer of acetyl groups into mitochondria [53]. Severe alterations of glucose and fatty acids’ metabolism have previously been documented in SMA patients [54,55,56] and were associated with dysregulated energy homeostasis and fatigue [57]. Poor tolerance for fasting has also been observed in SMA patients who develop severe hypoglycemia and ketoacidosis [53,57,58,59]. Our results indicate that SMN upregulation induced by Nusinersen may directly address these metabolic defects by boosting glucose metabolism and, ultimately, ATP production in severe SMA1 patients. Further analyses are needed to determine whether enteral feeding (gastrostomy) might have contributed to the Nusinersen-dependent metabolic changes in SMA1 patients.

Our analysis identified modulation of energy-related biochemical pathways in the CSF of SMA2 patients treated with Nusinersen relative to the baseline (Figure 4). However, these metabolic effects are different from those in SMA1 patients because Nusinersen does not affect glucose and pyruvic acid in the CSF of SMA2 patients. In contrast, Nusinersen modulates ketone body metabolism, as indicated by increased CSF levels of acetone, acetoacetate, and 3-hydroxybutyrate. In addition to being the primary energy sources for the CNS under fasting conditions [60,61], ketone bodies can buffer hydroxyl radicals and inhibit ROS production [62], thereby contributing to the maintenance of redox homeostasis in neurons. It remains to be clarified whether the induction of ketone body metabolism in the CSF of Nusinersen-treated SMA2 patients might be influenced by the fasting period that precedes the lumbar puncture, which differs between SMA1 (<4 h) and SMA2 and SMA3 (>6 h) patients. However, this is unlikely because Nusinersen does not affect ketone body metabolism in the CSF of SMA3 patients who undergo the same fasting period as SMA2 patients.

The metabolic effects of Nusinersen in the CSF of SMA3 patients markedly differ from those in more severe SMA patients (Figure 5). Indeed, the only biochemical changes are linked to the metabolism of amino acids. Importantly, Nusinersen-dependent modulation of amino acid metabolism is common, despite distinctive features, for all SMA patients, irrespective of clinical severity. Considering the role of amino acids as crucial intermediates of diverse biochemical pathways [63], it is conceivable that the modulation of amino acid metabolism induced by Nusinersen contributes to drive the other metabolic changes that are specific to each SMA type. 

Knowledge of therapy-dependent biochemical changes and their correlation to clinical severity is currently limited. We began to address this issue by investigating the potential relationship of neurometabolic changes induced by Nusinersen with disease severity. Interestingly, hierarchical clustering analysis of neurometabolic changes induced by Nusinersen in the CSF of SMA1 patients, relative to untreated SMA1 and SMA2 patients, revealed a closer similarity of the biochemical profile of treated SMA1 patients at the maintenance phase with that of SMA2 patients at the baseline (Figure 6A). Similarly, a closer relationship emerged between the CSF metabolome of treated SMA2 patients at T2 and that of less severe SMA3 patients under naive conditions, compared to than that of untreated SMA2 individuals (Figure 6B). These results provide an initial biochemical framework to decipher the beneficial therapeutic effects of Nusinersen in the CSF of SMA patients at the metabolic level.

Our findings provide the first evidence for system-wide modulation of peripheral organ metabolism induced by intrathecal administration of Nusinersen and CNS-specific upregulation of SMN. In SMA1 patients, this possibility is supported by Nusinersen’s stimulation of biochemical pathways, such as the glucose-alanine cycle that directly involves liver and muscle function [52,64]. Similarly, the modulation of ketone body metabolism and fatty acids’ biosynthesis in SMA2 patients suggest activity of the liver, skeletal muscle, and adipose tissue [61,65]. Furthermore, increased ammonia recycling in SMA3 patients suggests Nusinersen-dependent modulation of liver function [65]. System-wide metabolic imbalance and dysfunction of several peripheral organs have been documented in mouse models of the disease and in SMA patients [56,66]. Our results are consistent with a scenario in which Nusinersen therapy modulates the peripheral metabolism in SMA patients by activating the metabolic regulatory loops involving the functional interaction between CNS and peripheral organs. Future studies, including the metabolomic analysis of blood serum in larger cohorts of SMA patients, are needed to corroborate this conclusion.

What are the mechanisms underlying the complex neurometabolic effects of Nusinersen treatment in SMA patients? A Nusinersen-dependent increase in SMN levels in the CNS is conceivably the upstream trigger for these events [12,67]. Downstream of SMN induction, the effects of Nusinersen treatment on tyrosine, phenylalanine, tryptophan, glutamine, serine, and histidine metabolism, across different SMA types, suggest a role for changes in CNS neurotransmission. Accordingly, these amino acids are directly involved in the synthesis of catecholamines (dopamine, and noradrenaline), serotonin, histamine, and excitatory neurotransmitters (glutamate, aspartate, glycine, and D-serine). Therefore, it could be envisioned that Nusinersen might stimulate neurotransmitter metabolism and directly influence the neuromuscular system through the functional improvement of motor neuron connections with skeletal muscle, thereby triggering further interplay with the liver and adipose tissue metabolism. Additionally, changes in central neurotransmission could influence systemic metabolism through regulation of the hypothalamus–pituitary axis, followed by the downstream hormonal modulation of peripheral organ metabolism in treated SMA patients.

Our NMR-based CSF analysis after Nusinersen treatment reveals a prominent effect on amino acid metabolism as well as the potential risk of inducing a harmful deficiency of these vital biomolecules, especially in severe SMA patients where a metabolic boost in the consumption of essential amino acids for fueling bioenergetic pathways is most apparent. Therefore, we suggest that dietary supplementation of specific amino acids could be tailored to potentiate the therapeutic effect of Nusinersen and counteract the deleterious effects of potentially depleting essential or conditional amino acids in treated SMA patients.

In conclusion, our findings highlight the remarkable effects of Nusinersen in regulating diverse biochemical pathways with unique signatures that are SMA-type-specific. Moreover, the modulation of amino acid metabolism together with the profound stimulation of bioenergetic pathways suggest that this CNS-directed therapy promotes a functional interplay involving muscle and peripheral organ metabolism in SMA patients. 

## Figures and Tables

**Figure 1 biomolecules-12-01431-f001:**
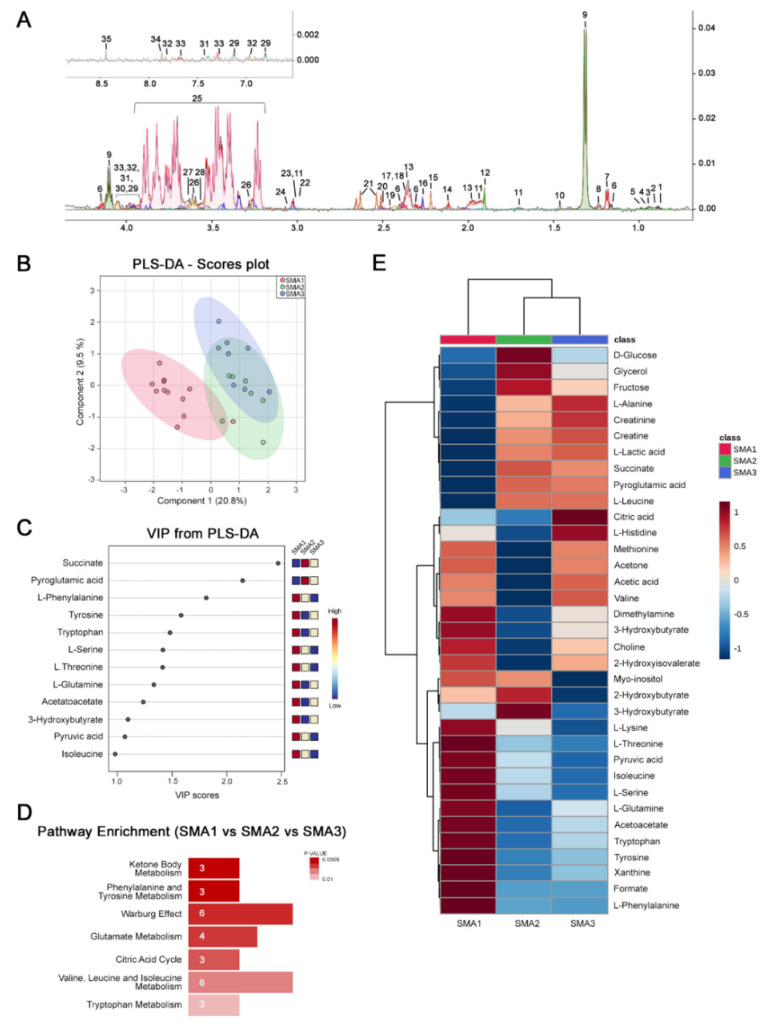
**Metabolomic profile of CSF from naive SMA1, SMA2, and SMA3 patients**. (**A**) Representative 1D ^1^H CPMG NMR spectrum obtained from the CSF of one SMA1 patient. CSF polar extracts of SMA patients detected the presence of 35 metabolites (1: 2-Hydroxybutyric acid; 2: 2-Hydroxyisovalerate; 3: L-leucine; 4: L-isoleucine; 5: L-valine; 6: 3-Hydroxybutyric acid; 7: 3-Hydroxyisobutyrate; 8: L-threonine; 9: lactic acid; 10: L-alanine; 11: L-lysine; 12: acetic acid; 13: L-glutamine; 14: L-methionine; 15: acetone; 16: acetoacetate; 17: pyroglutamic acid; 18: succinic acid; 19: pyruvic acid; 20: dimethylamine; 21: citric acid; 22: creatine; 23: creatinine; 24: choline; 25: D-glucose; 26: myo-inositol; 27: fructose; 28: glycerol; 29: L-tyrosine; 30: L-serine; 31: L-phenylalanine; 32: L-histidine; 33: L-tryptophan; 34: xantine; 35: formic acid). (**B**) PLS-DA score scatter plots showing the metabolomic profile of CSF from SMA1, SMA 2, and SMA3 patients prior to treatment. The cluster analyses are reported in the Cartesian space, which is described by the main components, PC1: 20.8% and PC2: 9.5%. PLS-DA was evaluated using cross-validation (CV) analysis. CV tests performed according to PLS-DA statistical protocol show a significant separation between SMA1, SMA2, and SMA3 at T0 (0.44 and 0.69 accuracy values on PC1 and PC2, respectively, and positive 0.31 and 0.437 Q2 indexes). (**C**) VIP score graphs of the metabolites discriminating the CSF of SMA1, SMA2, and SMA3 patients at T0. Metabolites characterized by a VIP score > 1 are shown. (**D**) Diagram of the pathway enrichment analysis showing the most dysregulated pathways at baseline. The number of molecules (hits) related to the specific metabolic pathway is shown within each bar. (**E**) Hierarchical heat maps generated by MetaboAnalyst software are based on the Euclidean distance and Ward’s algorithm. The bar color represents each metabolite’s abundance on a normalized scale from blue (low level) to red (high level). The dendrogram at the top is based on the similarity of the metabolomic profile relative to each sample cluster. The dendrogram on the left is based on the metabolite abundance profiles.

**Figure 2 biomolecules-12-01431-f002:**
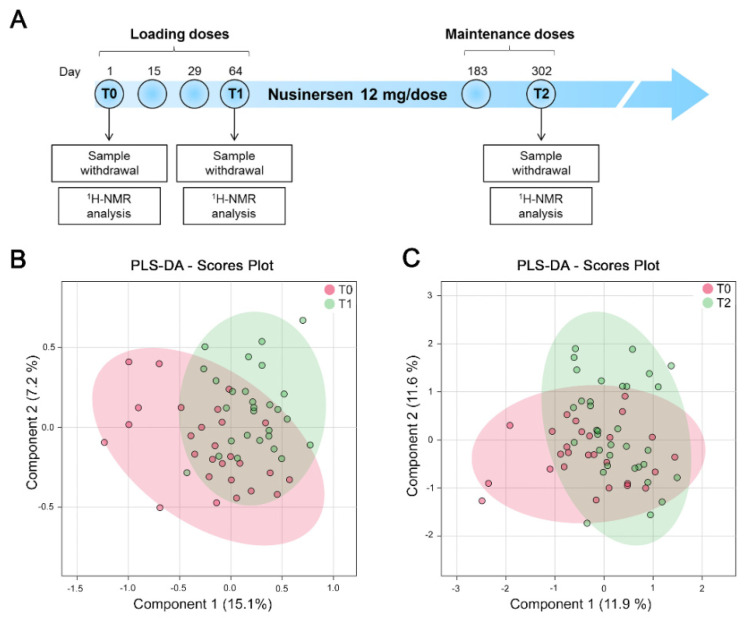
**NMR analysis does not identify Nusinersen-dependent neurometabolomic effects in the CSF of a pooled cohort of SMA patients with different disease severity.** (**A**) Schematic representation of the timeline of Nusinersen administration and CSF collection in SMA patients. (**B**,**C**) Multivariate statistical analysis performed on the cluster concentration matrices of the whole cohort of SMA1-3 patients collected before Nusinersen administration (T0) and at loading (T1) or maintenance phase (T2) using MetaboAnalyst 5.0. Multivariate statistical analysis produced PLS-DA score scatter plots relative to the CSF composition from SMA patients at T0 and T1 (**B**) as well as at T0 and T2 (**C**). This analysis revealed the absence of significant metabolomic differences in the CSF of the whole cohort of SMA patients either at T1 or T2, relative to untreated patients at T0. PLS-DA was evaluated using cross validation (CV) analysis. The clusters analyses are reported in the Cartesian space that is described by the main components, PC1: 15.1% and PC2: 7.2% (T0 vs. T1 clusters) (**B**) and PC1: 11.9% and PC2: 11.6% (T0 vs. T2 clusters) (**C**). CV reported Q2 negative values of −0.02 and −0.31 for the first and second principal component of SMA patients at T0 and T1, respectively; and of −0.44 and −0.84 for SMA patients at T0 and T2, respectively (see also Appendix A).

**Figure 3 biomolecules-12-01431-f003:**
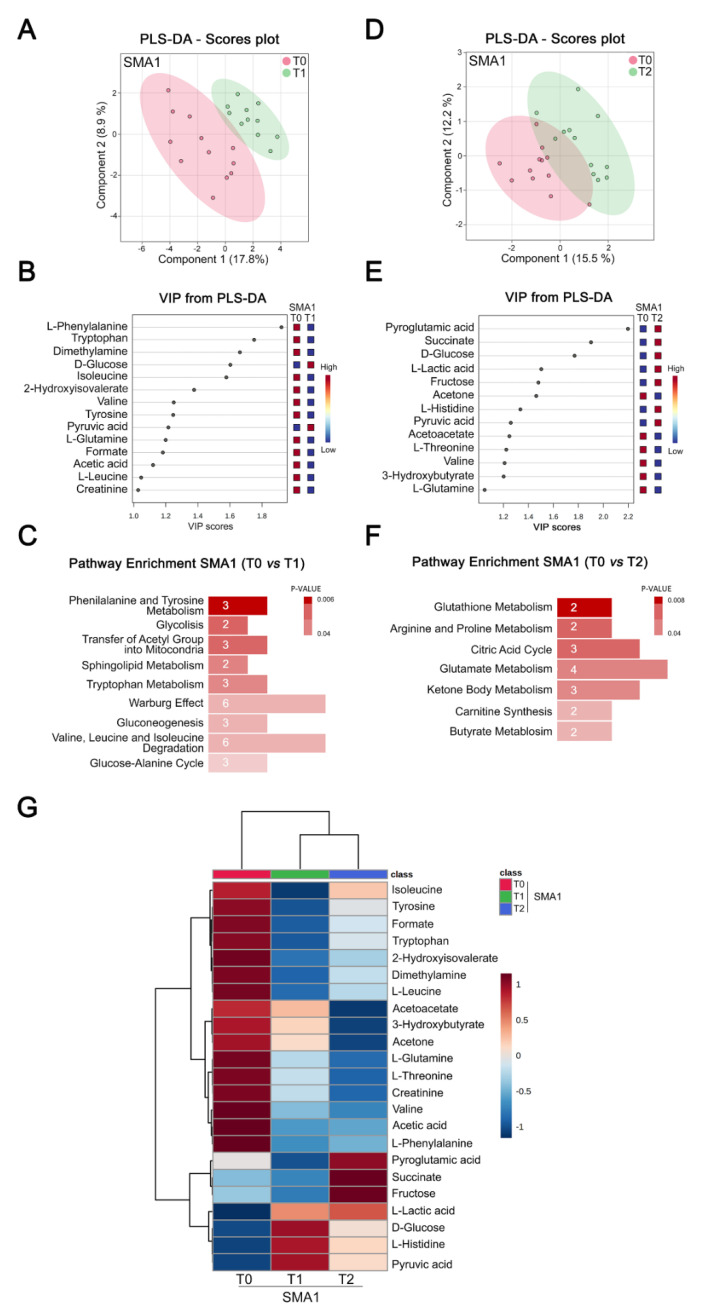
**Nusinersen modulates amino acid and glucose metabolism in the CSF of SMA1 patients.** (**A**,**D**) PLS-DA score scatter plots showing the metabolomic profiles of CSF for SMA1 patients prior to treatment (T0) and at loading (T1) (**A**) or maintenance (T2) phases (**D**) of Nusinersen administration. PLS-DA was evaluated using cross validation (CV) analysis. The clusters analyses are reported in the Cartesian space that is described by the main components, PC1: 17.8% and PC2: 8.9% (T0 vs. T1 clusters, respectively) (**A**), and PC1: 15.5% and PC2: 12.2% (T0 vs. T2 clusters, respectively) (**D**). CV tests performed according to PLS-DA statistical protocol show a significant separation between T0 and T1 clusters (0.68 and 0.77 accuracy values for PC1 and PC2, respectively, and positive 0.14 and 0.43 Q2 indices) (**A**), and T0 and T2 clusters (0.70 and 0.67 accuracy values for PC1 and PC2, respectively, and positive 0.02 and 0.03 Q2 indices) (**D**). The data reveal that Nusinersen induces metabolic changes in the CSF of SMA1 patients at both loading and maintenance phases. (**B**,**E**) VIP score graphs of the metabolites discriminating the CSF of SMA1 patients at T0 from those of the same patients at T1 (**B**) or T2 (**E**). Metabolites characterized by a VIP score > 1 are shown. (**C**,**F**) Diagram of the pathway enrichment analysis showing the effect of Nusinersen therapy on several biochemical pathways associated with amino acid and glucose metabolism in SMA1 patients at both T1 (**C**) and T2 (**F**). The number of molecules (hits) related to the specific metabolic pathway is shown within each bar (see also Appendix A). (**G**) Hierarchical heat maps generated by MetaboAnalyst software are based on the Euclidean distance and Ward’s algorithm. The heatmaps are calculated based on the concentrations of discriminating metabolites with VIP > 1 in the CSF of SMA1 patients at T0, T1, and T2. For each metabolite, the bar color represents its abundance on a normalized scale from blue (low level) to red (high level). The dendrogram at the top is based on the similarity of the metabolomic profile relative to each sample cluster. The dendrogram on the left is based on the metabolite abundance profiles. This analysis highlights hierarchical separation of CSF metabolomic profiles at loading (T1) and maintenance (T2) phases from the baseline (T0). This confirms that Nusinersen administration induces changes in the metabolomic CSF profile of SMA1 patients.

**Figure 4 biomolecules-12-01431-f004:**
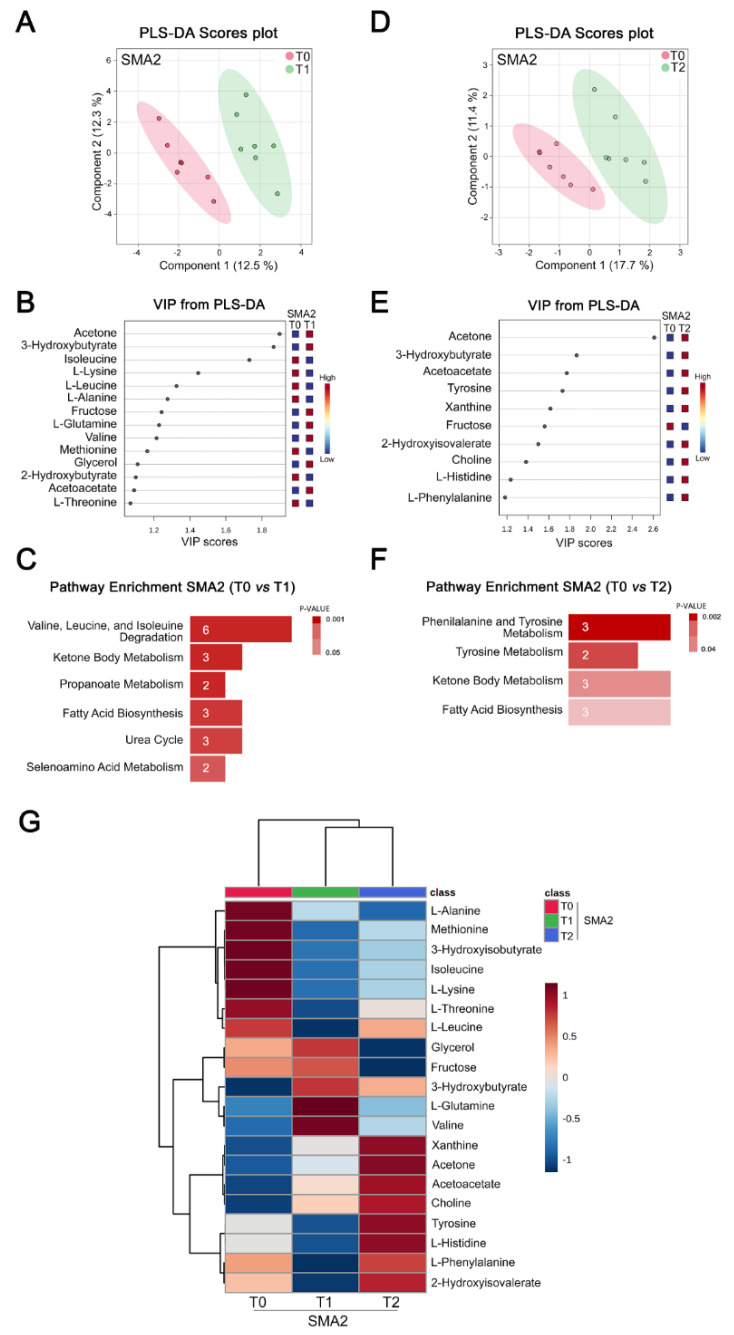
**Nusinersen modulates amino acid and ketone body metabolism in the CSF of SMA2 patients.** (**A**,**D**) PLS-DA score scatter plots showing the metabolomic profile of CSF from SMA2 patients before treatment (T0) and at loading (T1) (**A**) or maintenance (T2) (**D**) phases of Nusinersen administration. The clusters’ analyses are reported in the Cartesian space that is described by the main components, PC1: 12.5% and PC2: 12.3% (T0 vs. T1 clusters) (**A**) and PC1: 17.7% and PC2: 11.4% (T0 vs. T2 clusters) (**D**). PLS-DA was evaluated using cross validation (CV) analysis. CV tests performed according to PLS-DA statistical protocol show a significant separation between T0 and T1 clusters (0.94 and 1.0 accuracy values for PC1 and PC2, respectively, with positive 0.58 and 0.83 Q2 indices) (**A**), and T0 and T2 clusters (0.57 and 0.71 accuracy values for PC1 and PC2, respectively, and 0.23 and 0.29 Q2 indices) (**D**). The data reveal that Nusinersen induces metabolic changes in the CSF of SMA2 patients at both loading and maintenance phases. (**B**,**E**) VIP score graphs of metabolites discriminating the CSF of SMA2 patients at T0 from that of the same patients at T1 (**B**) or T2 (**E**). Metabolites characterized by a VIP score > 1 are shown. (**C**,**F**) Diagram of the pathway enrichment analysis showing the effect of Nusinersen therapy on biochemical pathways associated with amino acid and ketone body metabolism in SMA2 patients at both T1 (**C**) and T2 (**F**). The number of molecules (hits) related to the specific metabolic pathway is shown within each bar (see also Appendix A). (**G**) Hierarchical heat maps generated by MetaboAnalyst software are based on the Euclidean distance and Ward’s algorithm. The heatmaps are calculated based on the concentrations of discriminating metabolites with VIP > 1 in the CSF of SMA2 patients at T0, T1, and T2. For each metabolite, the bar color represents its abundance on a normalized scale from blue (low level) to red (high level). The dendrogram at the top is based on the similarity of the metabolomic profile relative to each sample cluster. The dendrogram on the left is based on the metabolite abundance profiles. The analysis highlights hierarchical separation of CSF metabolomic profiles at loading (T1) and maintenance (T2) phases from the baseline (T0). This confirms that Nusinersen administration induces changes in the metabolomic CSF profile of SMA2 patients.

**Figure 5 biomolecules-12-01431-f005:**
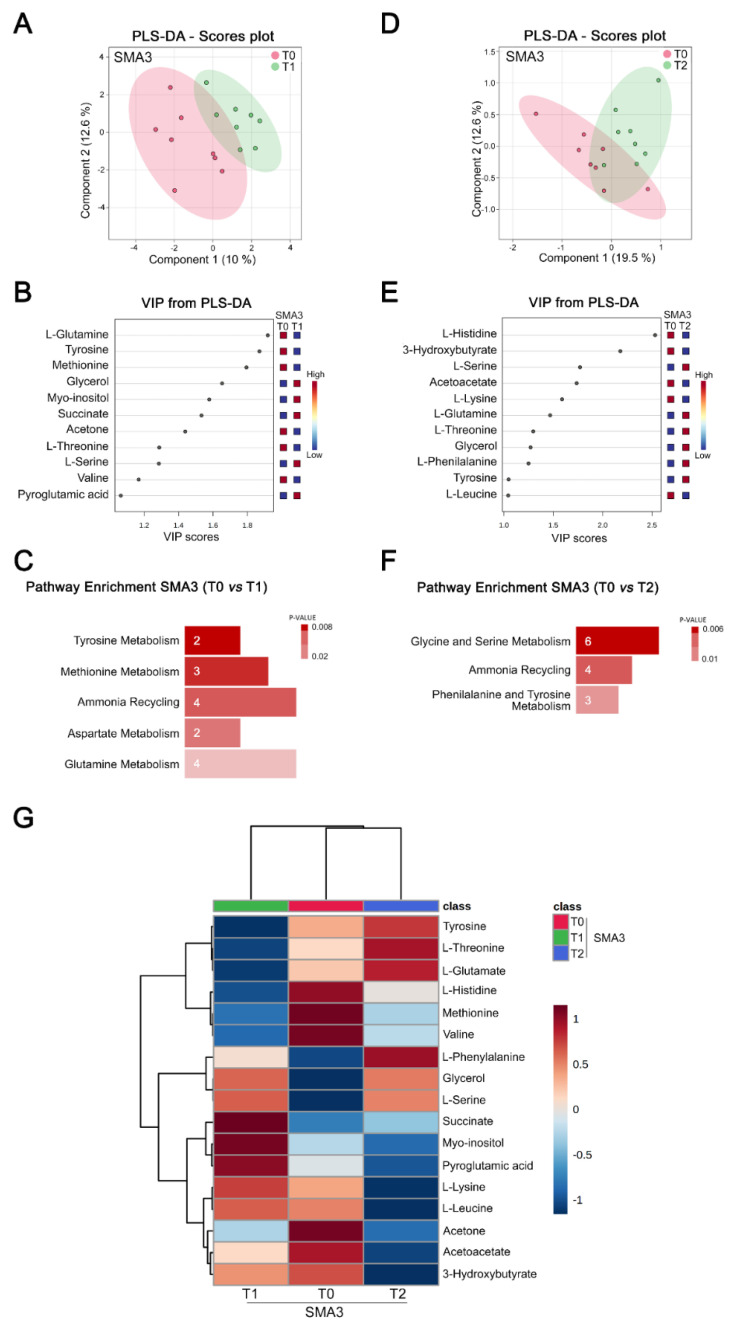
**Nusinersen modulates amino acid metabolism in the CSF of SMA3 patients.** (**A**,**D**) PLS-DA score scatter plots showing the metabolomic profile of CSF from SMA3 patients prior to treatment (T0) and at loading (T1) (**A**) or maintenance (T2) (**D**) phases of Nusinersen administration. The clusters analyses are reported in the Cartesian space that is described by the main components PC1: 10.0% and PC2: 12.6% (T0 vs. T1 clusters, respectively) (**A**) and PC1: 19.5% and PC2: 12.6% (T0 vs. T2 clusters, respectively) (**D**). PLS-DA was evaluated using cross validation (CV) analysis. CV tests performed according to PLS-DA statistical protocol show a significant separation between T0 and T1 clusters (0.68 and 0.77 accuracy values for PC1 and PC2, respectively, and positive 0.14 and 0.43 Q2 indexes) (**A**), and T0 and T2 clusters (0.74 and 0.76 accuracy values for PC1 and PC2, respectively, and positive 0.16 and 0.18 Q2 values) (**D**). The data reveal that Nusinersen induces metabolic changes in the CSF of SMA3 patients at both loading and maintenance phases. (**B**,**E**) VIP score graphs of the metabolites discriminating the CSF of SMA3 patients at T0 from that of the same patients at T1 (**B**) or T2 (**E**). Metabolites characterized by a VIP score > 1 are shown. (**C**,**F**) Diagram of the pathway enrichment analysis showing a selective effect of Nusinersen therapy on biochemical pathways associated with amino acid metabolism in SMA3 patients at both T1 (**C**) and T2 (**F**). The number of molecules (hits) related to the specific metabolic pathway is shown within each bar (see also Appendix A). (**G**) Hierarchical heat maps generated by MetaboAnalyst software are based on the Euclidean distance and Ward’s algorithm. The heatmaps are calculated based on the concentrations of discriminating metabolites with VIP > 1 in the CSF of SMA3 patients at T0, T1, and T2. For each metabolite, the bar color represents its abundance on a normalized scale from blue (low level) to red (high level). The dendrogram at the top is based on the similarity of the metabolomic profile relative to each sample cluster. The dendrogram on the left is based on the metabolite abundance profiles.

**Figure 6 biomolecules-12-01431-f006:**
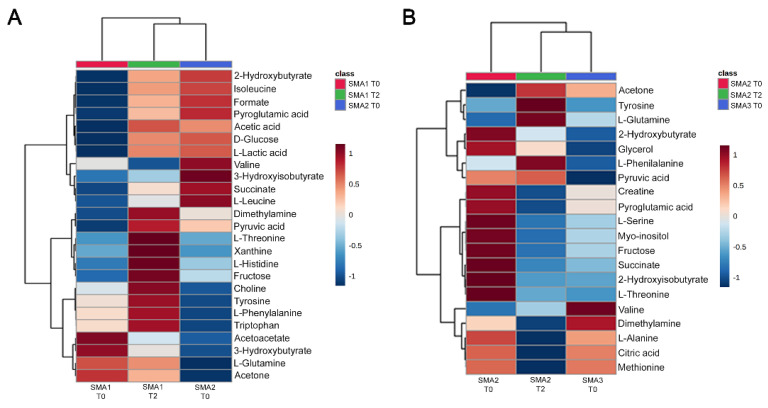
**Comparative analysis of Nusinersen-dependent neurometabolic changes and disease severity.** (**A**,**B**) Hierarchical clustering of metabolites identified by NMR analysis from the CSF of SMA1 (T0 and T2) and SMA2 (T0) patients (**A**) or SMA2 (T0 and T2) and SMA3 (T0) patients (**B**) shows that the metabolomic changes in treated SMA1 and SMA2 patients resemble the profile of untreated patients with a milder form of the disease. The heatmaps generated by MetaboAnalyst software are based on the Euclidean distance and Ward’s algorithm and are calculated based on the concentrations of discriminating metabolites with VIP > 1. For each metabolite, the bar color represents its abundance on a normalized scale from blue (low level) to red (high level). The dendrogram at the top is based on the similarity of the metabolomic profile relative to each sample cluster. The dendrogram on the left is based on the metabolite abundance profiles.

**Table 1 biomolecules-12-01431-t001:** Demographic and clinical characteristics of SMA patients.

	All SMA		SMA1		SMA2		SMA3	
	*N*	*Median [Min; Max]*	*N*	*Median [Min; Max]*	*N*	*Median [Min; Max]*	*N*	*Median [Min; Max]*
*Number*	27		12		7		8	
*Gender (female/male)*	19/8		8/4		6/1		5/3	
*SMN2 copies (2/3/4)*	13/13/1		11/1/0		0/7/0		2/5/1	
*Age (years)*	27	5.05 [0.57; 17.95]	12	3.91 [0.57; 7.78] #	7	5.49 [1.44; 12.53]	8	12.83 [2.3; 17.95]
*Height (m)*	26	1.12 [0.68; 1.67]	12	1.04 [0.68; 1.27]	7	1.11 [0.83; 1.58]	7	1.52 [0.8; 1.67]
*Weight (kg)*	26	16.3 [7.9; 68]	12	12.6 [7.9; 26] #	7	17.8 [10.7; 49]	7	42.00 [10.5; 68]
*Disease duration at therapy onset (years)*	27	3.87 [0; 15.13]	12	3.61 [0; 7.36]	7	4.14 [0.33; 11.53]	8	2.06 [0.34; 15.13]
*BMI*	26	15.91 [11.57; 26.56]	12	13.72 [11.57; 17.08] ##	7	17.57 [12.82; 19.63] §	7	19.44 [16.22; 26.56]
*CHOP-INTEND*	12	8.5 [0; 52]	12	8.5 [0; 52]	-		-	
*HFMSE*	15	16.00 [2; 61]	-		7	10 [2; 16] ##	8	37.5 [15; 61]
*Gastrostomy (no/yes)*	17/10		2/10		7/0		8/0	
*NIV (no/yes)*	19/8		8/4		4/3		7/1	
*Tracheostomy (no/yes)*	21/6		6/6		7/0		8/0	

# *p* < 0.05, ## *p* < 0.01, compared to SMA3 patients (Mann–Whitney test); § *p* < 0.05, compared to SMA1 patients (Mann–Whitney test).

## Data Availability

The data that support the findings of this study are available from the corresponding author, upon reasonable request.

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
