# Peer review of "Nusinersen Induces Disease-Severity-Specific Neurometabolic Effects in Spinal Muscular Atrophy"

_biomolecules, 2022, doi:10.3390/biom12101431_

Round 1
Reviewer 1 Report
This is a very interesting study on the effect of nusinersen on the metabolism of SMA patients. In detail, the authors did not find metabolic variations considering the whole population of 27 patients, but specific effects by analyzing the different subgroups of type 1, type 2 and type 3. The patient subgroups are quite small (type 1 = 12, type 2 = 7, type 3 = 8), but the study finds significant changes, especially in type 1.
These findings may suggest that, within the same condition called SMA, we can find different metabolic imbalances related to the severity of the disease. Metabolic changes may play a role as biomarkers to define the patient's underlying condition and monitor drug effects. Other potential implications relate to nutritional supplementation (and treatment) in SMA patients.
Patient subgroups were divided according to the type of disease (1, 2 and 3). However, within the three subgroups, patients show a fairly wide range of clinical phenotypes, perhaps with partial overlap between subgroups (as suggested by the range of CHOP and HFMSE).
Do the authors believe that the correlation is stronger with the type of disease at baseline than with the functional status of the patient? If possible, add a comment in the discussion.
Do the authors think that enteral feeding via gastrostomy, present only in type 1, may have influenced the results by modifying the metabolism of these patients? Please add a comment in the discussion.
Author Response
This is a very interesting study on the effect of nusinersen on the metabolism of SMA patients. In detail, the authors did not find metabolic variations considering the whole population of 27 patients, but specific effects by analyzing the different subgroups of type 1, type 2 and type 3. The patient subgroups are quite small (type 1 = 12, type 2 = 7, type 3 = 8), but the study finds significant changes, especially in type 1.
These findings may suggest that, within the same condition called SMA, we can find different metabolic imbalances related to the severity of the disease. Metabolic changes may play a role as biomarkers to define the patient's underlying condition and monitor drug effects. Other potential implications relate to nutritional supplementation (and treatment) in SMA patients.
We are pleased the Reviewer finds that this is a very interesting study with potential implications related to nutritional supplementation (and treatment) in SMA patients. We have addressed the specific points raised by the Reviewer as described below.
Patient subgroups were divided according to the type of disease (1, 2 and 3). However, within the three subgroups, patients show a fairly wide range of clinical phenotypes, perhaps with partial overlap between subgroups (as suggested by the range of CHOP and HFMSE).
1) Do the authors believe that the correlation is stronger with the type of disease at baseline than with the functional status of the patient? If possible, add a comment in the discussion.
R: We thank the Reviewer for their comment. Following the Reviewer’s cue, we have compared the basal CSF metabolomes of SMA1, SMA2 and SMA3 patients and found that a clear segregation of metabolomic profiles according to disease severity is already present prior to Nusinersen treatment. We have added the new data as Figure 1 in the revised manuscript and included appropriate description of these findings in both the Results and Discussion sections. We believe the new addition further improves our study by revealing baseline metabolomic differences as well as providing further elements to explain the effects of Nusinersen.
2) Do the authors think that enteral feeding via gastrostomy, present only in type 1, may have influenced the results by modifying the metabolism of these patients? Please add a comment in the discussion.
R: We thank the Reviewer for raising this point. As requested, we have now added a comment on this potential issue in the Discussion (page 16). Please note that due to the small and unbalanced number of SMA1 patients with (n=10) and without (n=2) gastrostomy, we could not further stratify our NMR-based data to evaluate the impact of this feeding support on the CSF metabolome. We would like to highlight that the enteral feeding formulas recommended for SMA are generally designed to meet the basic macro and micronutrient requirements of patients. Moreover, in our pediatric centers we carefully monitor in all type of SMA patients the nutritional status, the hypothetical rest energy expenditure, and the body composition in order to prescribe a proper diet. However, despite our patients undergoing well controlled nutritional conditions, we cannot exclude that enteral feeding via gastrostomy may affect the CSF metabolome.
Reviewer 2 Report
In their paper, the authors demonstrate that specific changes of CSF-metabolites (intermediates of bioenergetics pathways, amino acids, ketone bodies, lipids) occur in patients with SMA following the intrathecal administration of nusinersen, depending upon the SMA phenotype. The authors use NMR-spectroscopy to measure the individual compounds.
The methodology appears sound and the results are plausible. The paper is well written and the conclusions are relevant for the management of SMA patients.
There are, however, a few points to consider, which the authors should comment on:
1) Even within a particular SMA-group, there is considerable diversity of the clinical presentation, such as the presence/absence of a gastrostomy, non-invasive ventilation or HFMSE-scores. This clinical diversity can also be a parameter affecting the neurometabolic profiles
2) Does gender affect the neurometabolic profile?
3) The duration of fasting prior to CSF sampling was significantly different between the SMA1 (<4hrs) and SMA2 (>6hrs) patients. Although this is comprehensible in view of the age of the children, the duration of fasting will affect energy metabolism, i.e. glycolysis and ketone body degradation. This might partially explain the enhanced ketone body production which the authors observed in the SMA2-patients compared to the SMA1-patients
4) How do the authors explain the described systemic effects of nusinersen considering that the drug is not able to cross the blood-brain-barrier? It would be interesting to follow these changes not only in CSF, but also in blood.
5) As the severity of the SMA seems to determine the pattern of CSF metabolites, the analysis of biomarkers of neurodegeneration (CSF neurofilaments) would be helpful for a more detailed characterization of the correlation of the neurometabolic signature with the stage of the disease
Author Response
In their paper, the authors demonstrate that specific changes of CSF-metabolites (intermediates of bioenergetics pathways, amino acids, ketone bodies, lipids) occur in patients with SMA following the intrathecal administration of nusinersen, depending upon the SMA phenotype. The authors use NMR-spectroscopy to measure the individual compounds.
The methodology appears sound and the results are plausible. The paper is well written and the conclusions are relevant for the management of SMA patients.
There are, however, a few points to consider, which the authors should comment on:
We appreciate the Reviewer’s positive evaluation of our study and have addressed their specific comments as described below.
1) Even within a particular SMA-group, there is considerable diversity of the clinical presentation, such as the presence/absence of a gastrostomy, non-invasive ventilation or HFMSE-scores. This clinical diversity can also be a parameter affecting the neurometabolic profiles.
2) Does gender affect the neurometabolic profile?
R: We agree with the Reviewer about the clinical heterogeneity of our patients, which is to be expected considering the real-world nature of our analysis. Unfortunately, given the small sample size of our study, we could not further stratify the NMR-based metabolome data according to the presence/absence of gastrostomy or NIV, CHOP-INTEND or HFMSE scores, and gender. Therefore, as correctly stated by the Reviewer, it remains possible that these clinical and demographic features may have some influence on the CSF metabolome. We refer to future prospective studies on larger cohorts of SMA patients to address the role of these parameters on the CSF metabolome of SMA patients.
3) The duration of fasting prior to CSF sampling was significantly different between the SMA1 (<4hrs) and SMA2 (>6hrs) patients. Although this is comprehensible in view of the age of the children, the duration of fasting will affect energy metabolism, i.e. glycolysis and ketone body degradation. This might partially explain the enhanced ketone body production which the authors observed in the SMA2-patients compared to the SMA1-patients.
R: We thank the Reviewer for raising this point. Since fasting is known to induce hypoglycemia and enhance ketone body production, it may in principle differentially affect the neurometabolic profile of Nusinersen-treated SMA1 and SMA2 patients. However, to prevent hypoglycemia and ketoacidosis during fasting, we always administer intravenous hydration with glucose. We have now specified this procedure in the revised Materials and Methods (page 5). Furthermore, our NMR-based analysis revealed that Nusinersen is unable to alter ketone body metabolism of SMA3 patients, who undergo the same fasting as SMA2 patients. This makes it unlikely that increased ketone body production in SMA2 but not in SMA1 and SMA3 patients is directly linked to the duration of fasting. We have added a comment to address this possibility in the revised Discussion (see page 16).
4) How do the authors explain the described systemic effects of nusinersen considering that the drug is not able to cross the blood-brain-barrier? It would be interesting to follow these changes not only in CSF, but also in blood.
R: We agree with the Reviewer that the analysis of blood serum would be interesting. Unfortunately, blood samples from our cohort of SMA patients was not available for this retrospective study. We have added a comment about this limitation in the revised manuscript (see page 17).
5) As the severity of the SMA seems to determine the pattern of CSF metabolites, the analysis of biomarkers of neurodegeneration (CSF neurofilaments) would be helpful for a more detailed characterization of the correlation of the neurometabolic signature with the stage of the disease
R: We agree with the Reviewer that the analysis of CSF would have allowed us to further correlate the neurometabolomic profile of SMA patients with the clinical severity of SMA. However, the analysis of neurofilaments was not available to us, and thus cannot be included in this retrospective study.
Reviewer 3 Report
Errico and colleagues present a manuscript in which CSF is profiled from SMA1, SMA2 and SMA 3 patients at baseline and after treatments with Nusiresen. Focusing on metabolites, the Authors present some interesting findings. In particular, in severe SMA1 patients, Nusinersen stimulates energy-related glucose metabolism; in SMA2 patients, Nusinersen effects are related to energy homeostasis but involve ketone body and fatty acid biosynthesis; in SMA3 patients, Nusinersen mainly modulates amino acid metabolism. The article is well written and provides a good addition to the field. However, before publication I have a few suggested changes that the authors should consider:
- The Authors should include other clinical data in the table 1, specifically: disease duration at therapy onset, SMA maximum motor function at baseline; motor function score after treatment at T2.
-To test whether the identified metabolites could be related to different clinical response to therapy, the Authors should perform a regression analysis among CSF metabolite levels with: HFMSE score at baseline and after treatment, and between metabolite levels with ΔHFMSE (i.e., HFMSE score post treatment – HFMSE pre-treatment); CHOP INTEND; SMN2 copy number.
- Have the Authors an idea of ​​what happens in the serum of SMA patients in term of metabolites? More ambitiously, they could perform the metabolite analysis in the subgroup of SMA patients (the more severe form) to strengthen their hypothesis on the CNS-directed therapy that promotes a functional interplay involving muscle and peripheral organ metabolism.
Author Response
Errico and colleagues present a manuscript in which CSF is profiled from SMA1, SMA2 and SMA3 patients at baseline and after treatments with Nusiresen. Focusing on metabolites, the Authors present some interesting findings. In particular, in severe SMA1 patients, Nusinersen stimulates energy-related glucose metabolism; in SMA2 patients, Nusinersen effects are related to energy homeostasis but involve ketone body and fatty acid biosynthesis; in SMA3 patients, Nusinersen mainly modulates amino acid metabolism. The article is well written and provides a good addition to the field. However, before publication I have a few suggested changes that the authors should consider:
We appreciate the Reviewer’s overall comment on the quality and relevance of our study for the field. We have addressed their specific comments as described below.
1) The Authors should include other clinical data in the table 1, specifically: disease duration at therapy onset, SMA maximum motor function at baseline; motor function score after treatment at T2.
R: We thank the Reviewer for the suggestion. We have now added new data that further clarify the clinical features of SMA patients (see the new version of Table 1).
2) To test whether the identified metabolites could be related to different clinical response to therapy, the Authors should perform a regression analysis among CSF metabolite levels with: HFMSE score at baseline and after treatment, and between metabolite levels with ΔHFMSE (i.e., HFMSE score post treatment – HFMSE pre-treatment); CHOP INTEND; SMN2 copy number.
R: We thank the Reviewer for giving us the chance to perform new statistical analyses that have enriched the comprehension of the complex interactions between the neurometabolic profile of SMA patients and their clinical outcomes. Based on this new information, we have now added a new Supplementary Table (Supplementary Table S5) and modified the text accordingly (see Materials and Methods, page 6; Results, page 15).
3) Have the Authors an idea of ​​what happens in the serum of SMA patients in term of metabolites? More ambitiously, they could perform the metabolite analysis in the subgroup of SMA patients (the more severe form) to strengthen their hypothesis on the CNS-directed therapy that promotes a functional interplay involving muscle and peripheral organ metabolism.
R: We acknowledge the relevance of this question, which was also raised by Reviewer #2. Unfortunately, blood samples from our cohort of SMA patients was not available for this retrospective study. We refer to future prospective studies to further support our hypothesis of a functional interplay between central and peripheral systems of SMA patients. We have added a comment about this limitation in the revised manuscript (see page 17).
Round 2
Reviewer 3 Report
The authors have fully replied to all the issues made, the paper is now ready for publication